# Key role of Extracellular RNA in hypoxic stress induced myocardial injury

**Saumya Bhagat[1], Indranil Biswas[1], Md Iqbal Alam[2], Madiha Khan[3], Gausal A. Khan[4]***

**1** Department of Physiology, Defence Institute of Physiology and Allied Sciences, Timarpur, New Delhi, India, **2** Department of Physiology, HIMSAR, Jamia Hamdard, Hamdard Nagar, New Delhi, India, **3** The Heritage School, Kolkata, India, **4** Department of Physiology & Physiotherapy, College of Medicine, Nursing & Health Sciences, Fiji National University, Suva, Fiji Islands

* gausalk@gmail.com

**Data Availability Statement:** All relevant data are within the paper and its Supporting Information files.

**Funding:** This work was supported by grants from the Ministry of Defense (DIP253), Govt. of India

## Abstract

Myocardial infarction (MI), atherosclerosis and other inflammatory and ischemic cardiovascular diseases (CVDs) have a very high mortality rate and limited therapeutic options. Although the diagnosis is based on markers such as cardiac Troponin-T (cTrop-T), the mechanism of cTrop-T upregulation and release is relatively obscure. In the present study, we have investigated the mechanism of cTrop-T release during acute hypoxia (AH) in a mice model by ELISA & immunohistochemistry. Our study showed that AH exposure significantly induces the expression and release of sterile inflammatory as well as MI markers in a time-dependent manner. We further demonstrated that activation of TLR3 (mediated by eRNA) by AH exposure in mice induced cTrop-T release and Poly I:C (TLR3 agonist) also induced cTrop-T release, but the pre-treatment of TLR3 immuno-neutralizing antibody or silencing of *Tlr3* gene or RNaseA treatment two hrs before AH exposure, significantly abrogated AH-induced Caspase 3 activity as well as cTrop-T release. Our immunohistochemistry and Masson Trichrome (MT) staining studies further established the progression of myocardial injury by collagen accumulation, endothelial cell and leukocyte activation and adhesion in myocardial tissue which was abrogated significantly by pre-treatment of RNaseA 2 hrs before AH exposure. These data indicate that AH induced cTrop-T release is mediated via the eRNA-TLR3-Caspase 3 pathway.

## Introduction

Ascent to high altitudes generates a hypoxic environment in the body, causing hypoxemia in the circulation that leads to inflammation and hypercoagulation [1]. Several disorders such as CVDs, pulmonary and cerebral edema are concomitant with hypoxia-induced inflammation [2]. The fundamental cause of morbidity and mortality in MI [3] and acute lung injury [4] has been observed to be vascular inflammation due to hypoxia.

Clinically, acute myocardial infarction (AMI) is a condition wherein cTrop-T concentration is elevated. Traditional diagnostic checks for AMI include an onset of chest pain and abnormalities in electrocardiographic (ECG) tests. These are either often absent and/or

and Grand Challenge Canada (R-ST-POC-1807-13914) to GAK. The funders had no role in study design, data collection and analysis, decision to publish, or preparation of the manuscript.

**Competing interests:** The authors have declared that no competing interests exist.

**Abbreviations:** AH, acute hypoxia; cTrop-T, cardiac troponin T; eRNA, extracellular RNA; MI, myocardial infarction; SI, sterile inflammation; TLR3, Toll-like receptor 3.

nonspecific. Therefore, the diagnosis is mainly dependent upon the elevated levels of cTrop-T which is a reliable AMI marker. It is well known that more than 90% of intracellular Troponin are localised in the sarcomere and the rest circulate in the cytoplasm. Therefore, the release of cTrop-T into the circulation followed by the events of myocyte necrosis, apoptosis, formation and release of membranous blebs, increases membrane permeability and release of proteolytic troponin degradation products [5].

Hypoxia induces stress which stimulates "sterile (non-pathogenic) inflammation". In response to such a stress, a synchronised inflammatory response repairs and/or eliminates the damaged cells, further aggravating the injury. This response is facilitated by the cellular components of the innate immune system (neutrophils, monocytes, macrophages, etc.) and also endothelial cells (ECs), fibroblasts and pericytes, which recognizes various Damage Associated Molecular Patterns (DAMPs) molecules that are inappropriately discharged from dying cells [6,7]. SI induced activation of the innate immune system plays a significant role in the genesis of several chronic diseases viz. atherosclerosis, heart failure, etc [8].

Studies have revealed that post-hypoxic exposure cardiac injury may be reversible (except for papillary muscle necrosis and infarction) [9]; although, cardiomyocyte injury occurs, even if reperfusion therapy is given as soon as possible following the hypoxic insult [10]. Tissue injury due to microbial pathogens or sterile stress such as oxygen free radicals, hypoxia, or hyperthermia upregulates various DAMPs, which may be released into the circulation or can remain at the site of injury and trigger the inflammatory response [11]. Furthermore, DAMPs also trigger and engage the cells of the innate immune system in the heart [12].

Extracellular RNA (eRNA), a DAMP, released during hypoxia causes inflammation and vascular diseases [13]. However, its role in hypoxia induced AMI is not fully understood. eRNA is predominantly derived from ischemic myocardium and plays an adverse role in myocardial ischemia [11]. Apoptosis and necrosis of the tissues release endogenous nucleic acids which are recognized by TLR3,7,8 & 9 and are vital response to viral infection [14]. Previous study in TLR3 deficient mice that were subjected to cardiac I/R have shown reduction in infarct size [15]. The vaso-protective role of RNaseA administration has been studied in cardioprotection for myocardial I/R (ischemia-reperfusion) injury in mice and in the isolated I/R Langendorff-perfused rat heart [16].

Given these links between hypoxia and AMI, we hypothesized that the diminished oxygen level stimulates eRNA release and subsequently triggers AMI through TLR3-caspase-3 pathway. Herein, we presented a possible mechanism of cTrop-T release and the progression of AMI in a murine model of AH. So, this paper assesses (i) the influence of hypoxia on the generation of SI molecules and TLR3 activation and (ii) the implication of TLR3 activation due to the release of cTrop-T and Caspase-3.

## Materials and methods

### Ethical clearance

Experimental protocols were approved by the Internal Review Board of Defence Institute of Physiology and Allied Sciences (DIPAS) (IAEC/DIPAS/2015-03; authorization no:27/GO/RBi/SL/99/CPCSEA) in accordance with the guidelines of the committee for the purpose of control and supervision of experiments on animal, Ministry of environment and forest, Government of India. This study was done on adult Swiss Albino mice weighing 25-30g.The mice were housed in the animal facility of the institute and provided with chow diet and water *ad libitum*. All experimental procedures have been carried out in DIPAS and animal procedures were performed under urethane (1.2g/kg) induced general anaesthesia to minimize suffering [1].

## Chemicals

The commercial details of the materials used for all experiments are as follows: Mouse monoclonal Troponin-T antibody (ab8295), Rabbit polyclonal Neutrophil Elastase antibody (ab21595), Mouse monoclonal PECAM-1 antibody (ab24590), Mouse monoclonal CD11b+CD18 antibody (ab13219), Mouse monoclonal CD41 antibody (ab11024), TLR3 neutralizing antibody (ab12085) and Rabbit polyclonal alpha smooth muscle Actin antibody (ab5694) were procured from Abcam (MA, USA). Mouse monoclonal Myoglobin Antibody (sc74525) was procured from Santa Cruz Biotechnology (TX, USA). RNaseA, DNase1, TLR3 siRNA sense (5′-CGUUA UCACACACCAUUUA-3′) and antisense (5′-UAAAUGGUGUGUGAUAACG-3′), negative-control siRNA (scrambled sequence) and Trichrome stain (Masson) kit (HT15) were procured from Sigma-Aldrich (MO, USA); MaxSuppressor In Vivo RNA-LANCEr II, a formulation that enables highly efficient siRNA delivery into animals was purchased from Bio Scientific Corporation (TX, USA). HRP conjugated secondary antibodies [anti-mouse (62–6520), anti-rabbit (65–6120), anti-goat (81–1620)], Alexa Fluor 488-conjugated anti-mouse (A-10680) secondary antibody and Rabbit polyclonal s100b antibody (PA5-78161) were procured from Invitrogen (CA, USA). Trizol (15596026), DAPI (D1306) and poly I:C (20148E) were procured from Thermo Fischer (MA, USA). Caspase-3 assay kit (K105-100) was purchased from BioVision Inc (CA, USA). Quantitative ELISA kits were procured as follows: Mouse HMGB1 protein ELISA kit (MBS722248) and Mouse vWF ELISA kit (MBS260427) from MyBiosource Inc (CA, USA); HSP70 high sensitivity ELISA kit (ADI-EKS-715) from Enzo Life Sciences (NY, USA); Mouse HSP90 ELISA kit (NBP2-76449) from Novus Biologicals (CO, USA); Mouse S100B ELISA kit (E-EL-M1033) from Elabscience (TX, USA).

## AH and other treatments

The mice were exposed to AH in an animal decompression chamber with conditions equivalent to atmospheric conditions at an altitude of 7628 m (282 mm Hg), with an $O_2$ content of ~8.5%, and standard temperature and humidity, as previously described [13,17]. This setting for hypoxia exposure has been previously standardized in our lab for research purposes [1,13,21,22]. As per experimental requirement, the durations of exposure to mice to hypobaric hypoxia are 0 hr, 6 hrs, 12 hrs, 24 hrs, 3 days and 7 days. Intravenous injection of RNaseA (1 mg/kg BW (body weight)) and DNase1 (5 mg/kg BW) [1,18,19] were done prior to AH exposure and blood was extracted post-exposure. The intravenous injection was performed with a 30-gauge needle insulin syringe (BD Biosciences). *In vivo* siRNA delivery was carried out as per protocols described elsewhere [13,20]. Poly I:C (TLR3 agonist) (1 mg/kg BW), TLR3 neutralizing (nt)-antibody (40 mg/kg BW) [21], Lipoteichoic Acid (TLR2 agonist) (100 μg/kg BW) [22] and HMGB1 nt-antibody (200 μg/kg BW) [22] were also administered intravenously [13,23]. 80 μg/kg BW of eRNA and eDNA (isolated from plasma of exposed animals, and purified by DNase1 and RNaseA treatment respectively) were injected into the animal intravenously through the tail vein six hrs prior to the sacrifice [1]. Plasma samples from hypoxia exposed mice were pooled to get better yield of eRNA and eDNA for further experiments. Sacrifice of control and treated animals was done by cervical dislocation by trained professional prior to harvest of organs and tissues for further studies [1].

## Blood collection and tissue harvest

Retro-orbital blood collection was done in 2 mL citrate vials from non-anesthetized mice [24]. The plasma was separated and stored for further experiments. The whole heart was excised from the sacrificed mice and stored separately for molecular work as well as tissue sectioning for further experiments and studies.

**eRNA and eDNA isolation** eRNA and eDNA were isolated from the plasma samples of hypoxia exposed mice by the Trizol method and a commercially available kit (Quick-cfDNA, Zymo Research) respectively, and stored after purification by DNase1/RNaseA treatment.

## Enzyme linked immunosorbent assay

ELISA was performed as previously described [25]. Briefly, 50μg total protein from plasma or tissue lysate samples was incubated overnight with an equal volume of coating buffer (0.5 M carbonate buffer [pH 9.6]) in an assay plate at 4˚C. Non-specific sites were blocked by BSA followed by incubation with primary antibodies for 2hrs (HMGB1, vWF, HSP70, HSP90, s100b, c-Trop T and Myoglobin at 1:3000 dilution in PBS buffer) and then specific HRP-conjugated secondary antibodies for 2hrs (anti-mouse, anti-rabbit and anti-goat secondary antibodies at 1:5000 dilution in PBS buffer). Protein detection was done using OPD substrate and color intensity absorbance was measured at 450 nm.

## *In vivo* gene silencing by TLR3 siRNA

*In vivo* TLR3 inhibition was done by intravenous injection of TLR3 siRNA using the MaxSuppressor kit as per manufacturer protocol as well as our previous studies [1,13]. Based on the delivery route (intravenous), the injection volume was 100 μl per dose. The injection was done intravenously into the mouse tail vein using an insulin syringe at 20 l/sec approximately. After the injection of the RNAi agent, the recommended incubation time in the animal is 3–4 days. Retro-orbital blood was drawn at the indicated times and the tissues were harvested for further processing.

## Histological assessment

Myocardial tissue was fixed in a formaldehyde solution and immersed in neutral buffered saline overnight. The tissues were processed for paraffin embedding and sectioning done using a standard microtome. Deparaffinization and rehydration of the section was done by xylene and alcohol gradient and the antigen retrieval was done by sodium citrate buffer. Endogenous peroxidase activity was blocked by treatment with 3% $H_2O_2$ and then permeabilized with 0.25% Triton X in PBS. After incubation of the sections with specific primary (α-SMA, PECAM-1 and NE at 1: 200 dilution) and secondary antibodies (anti-rabbit and anti-mouse at 1:500 dilution), the antibody binding sites were stained with DAB reagent and the images analyzed using ImageJ software (open-source imaging software), NIH.

For immunofluorescence microscopy, the sections were incubated overnight with primary antibodies (CD11/CD18 and CD41 at 1:250 dilution) at 4˚C. The sections were then washed thoroughly and incubated in the dark with Alexa 488 conjugated secondary antibodies (anti-mouse at 1:500 dilution) for 60 minutes at room temperature. The samples were treated with 0.01% DAPI for 15 minutes at room temperature for nuclear counterstaining. The sections were then viewed under a fluorescence microscope (Ti2-E Motorized Inverted Microscope; Nikon, Tokyo, Japan).

MT staining was done using a commercially available kit (HT15, Sigma Aldrich).

## Activated Caspase assay

The estimation of activated Caspase-3 was done using a commercial kit (Caspase-3 Fluorometric Assay Kit, BioVision Inc.), following the manufacturer's protocol.

### Statistical analysis

All experiments were performed in triplicate and the data expressed as means ± SEM. The statistical significance between experimental groups was determined by one-way ANOVA followed by Bonferroni's multiple comparison tests. A p-value of <0.05 was considered statistically significant.

## Results

### 1. Time dependent induction of SI molecules by AH

Time-dependent effect of AH exposure on the levels of circulating SI molecules was determined. Blood was collected from the mice exposed to AH for different time points (0–24 hrs) and the plasma was isolated. The mice that were subjected to AH exposure showed significantly higher levels of SI markers i.e., HMGB1, vWF, HSPs, s100b, eRNA and eDNA (Fig 1A–1G; p < 0.05). Thus, the six hrs exposure time remained constant for subsequent experiments unless otherwise stated.

### 2. AH induces myocardial injury in time dependant manner

To evaluate the time dependant effects of AH exposure on the myocardial injury, we analysed cardiac myocardial injury marker i.e., cTrop-T & myoglobin in plasma by ELISA (0–24 hrs and 3–7 days). It was observed that the mice that were subjected to AH exhibited a time dependent increase of cTrop-T and myoglobin when compared to the control until 7 days (Fig 2A and 2B; p < 0.05). This suggests that hypoxia exposure induced myocardial injury in a time dependent manner.

### 3. Contribution of eRNA in the AH induced cTrop-T liberation

The above results showed that AH exposure notably increased the circulating concentration of SI as well as cardiac injury markers. However, the involvement of particular sterile inflammatory molecules in the up-regulation of cTrop-T is obscure. Hence, we carried out treatment experiments on mice using RNaseA (RNA degrader) or DNase1 (DNA degrader) or HMGB1 neutralizing antibody or purified endogenous eRNA or eDNA [1], and the plasma level of cTrop-T was measured. Our analysis confirmed that the pre-treatment of RNaseA significantly decreased the plasma level of cTrop-T when compared to the control (Fig 3A; p<0.05). However, the pre-treatment of DNase1 /or HMGB1 neutralizing antibodies failed to do so (Fig 3B and 3C; p<0.05). Endogenous eRNA or eDNA is the positive control. These data suggest that eRNA is only involved in AH induced cTrop-T releases.

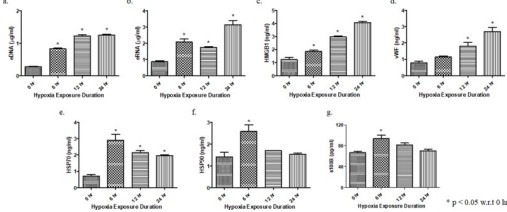

**Fig 1. AH induces release of circulating nucleic acids and expression of sterile inflammatory protein molecules.**
Level of SI markers in plasma was analysed after exposure to hypoxia for different time durations (0–24 hrs); (a) eDNA (μg/ml), (b) eRNA (μg/ml), (c) HMGB1 (ng/ml), (d) vWF (ng/ml), (e) HSP70 (ng/ml), (f) HSP90 (ng/ml), and (g) s100b (pg/ml). Data are shown as mean ± SEM (n = 5/group/each time point) from one experiment representative of three independent experiments, all performed in triplicate. One-way ANOVA revealed statistical significance in the results (*p< 0.05).

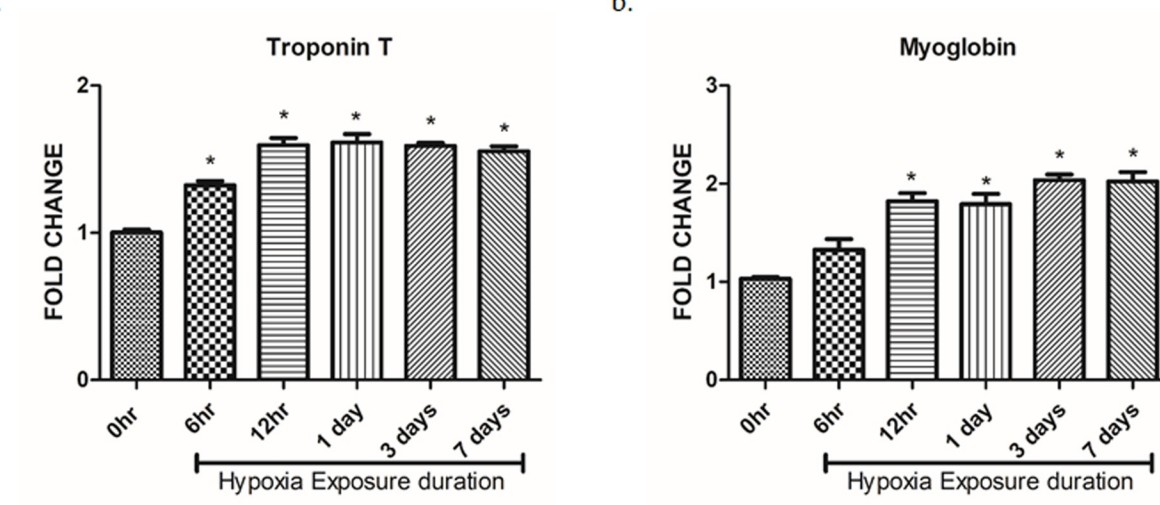

**Fig 2. AH exposure induces myocardial injury in time dependant manner.** Expression of cardiac injury markers in plasma samples after exposure of animals to hypoxia for different time durations (0–24 hrs, 3-7days); (a) cTrop-T, (b) Myoglobin. Data are shown as mean ± SEM (n = 5/group/treatment) from one experiment representative of three independent experiments, all performed in triplicate. One-way ANOVA revealed statistical significance in the results (*p< 0.05).

## 4. AH induced eRNA facilitates cTrop-T release via TLR3 signalling

Our earlier study showed that AH stimulated expression of TLR2 and TLR3 [22]. However, the involvement of TLRs in the hypoxia induced upregulation of cTrop-T is not clear. The above results revealed that the pre-treatment of HMGB1 neutralizing antibody failed to inhibit hypoxia induced cTrop-T release, also mice that were treated with LTA, a TLR2 agonist, failed to induce cTrop–T release. However, Poly I:C, a TLR3 agonist showed positive effect (induce cTrop–T release) (Fig 3D; p<0.05). This suggests the non-involvement of TLR2 in hypoxia induced upregulation of cTrop-T, which was similar to earlier results, showing that HMGB1 worked though TLR2 [22].

To further validate the association of TLR3 in release of cTrop-T under hypoxic condition, mice were treated with TLR3-immune neutralizing antibody, two hrs before exposure to AH, and the plasma cTrop-T was quantified by ELISA. Our result showed that pre-treatment of TLR3-immune neutralizing antibody significantly inhibited cTrop-T release (Fig 4A; p<0.05). To affirm the involvement of TLR3, we performed *Tlr3* gene silencing (70%) [1,13] *in vivo* by using *Tlr3* siRNA and the plasma level of cTrop-T was measured. Our results showed that AH-induced release of cTrop-T was substantially reduced in *Tlr3*-silenced mice as compared to the control (Fig 4B; p<0.05). This clearly demonstrates that hypoxia-induced release of cTrop-T is mediated through TLR3.

## 5. AH-induced eRNA enriched collagen accumulation, leukocyte infiltration and activation in cardiomyocytes

Vascular remodelling is one of the predominant factors to reduce the blood flow which consequently enhances hypoxic effect. To evaluate whether AH exposure has any role in vascular remodelling and subsequently myocardial injury, experiments were carried out to analyse expression of collagen and different markers for leukocyte activation in the cardiomyocytes. Fig 5A and 5B shows MT staining (for collagen) and immunohistochemistry

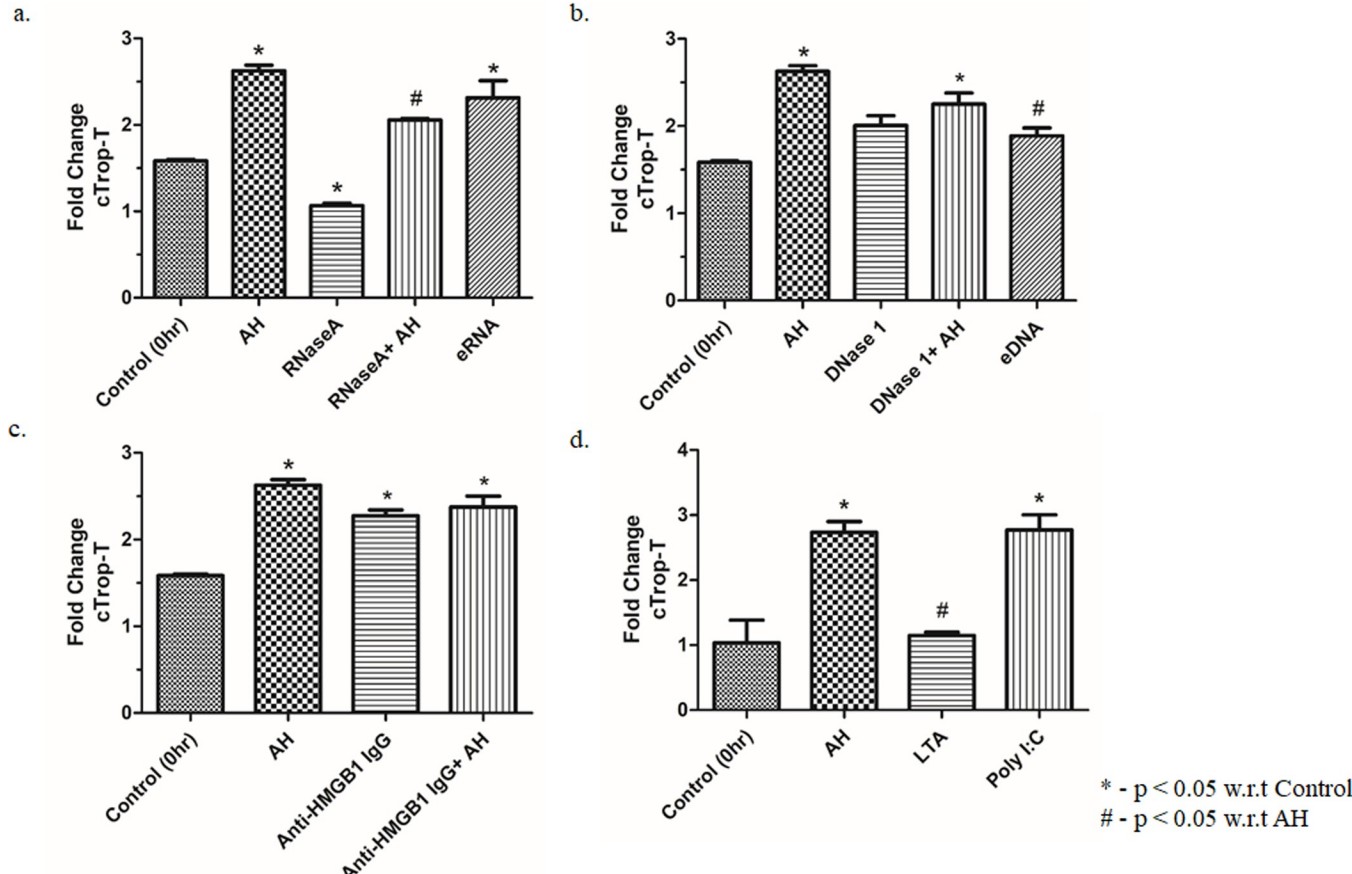

**Fig 3. eRNA is involved in hypoxia induced expression of cTrop-T.** ELISA estimation of circulating cTrop-T in plasma was done after (a) eRNA injection (80 μg/kg BW; 6 hrs prior to AH exposure) and RNaseA pre-treatment (1mg/kg BW; 2 hrs prior to AH exposure), (b) eDNA injection (80 μg/kg BW; 6 hrs prior to AH exposure) and DNase1 pre-treatment (5 mg/kg BW; 2 hrs prior to AH exposure), (c) HMGB1 IgG pre-treatment (200 μg/kg BW; 2 hrs prior to AH exposure), (d) pre-treatment with LTA (TLR2 agonist, 100 μg/kg BW; 2 hrs prior to AH exposure) and Poly I:C (TLR3 agonist, 1 mg/kg BW; 2 hrs prior to AH exposure). Control treatment with eRNA, eDNA, RNaseA, DNase 1, HMGB1 IgG, LTA and Poly I:C were done 6 hrs prior to sacrifice and the duration of AH exposure after different treatments was also 6 hrs. Data are shown as mean ± SEM (n = 5/group/treatment) from one experiment representative of three independent experiments, all performed in triplicate. One-way ANOVA revealed statistical significance in the results (*p< 0.05 groups w.r.t control; #p<0.05 groups w.r.t RNaseA).

staining for α-SMA (myofibroblast marker) in the myocardial tissue after various treatments, depicting collagen deposition in the peripheral vascular regions. Hypoxia-induced necrosis in myocardia initiates conversion of fibroblasts to myofibroblasts which requires increased synthesis of collagen, and this leads to its accumulation in myocardia. Collagen deposition and expression of α-SMA increases upon hypoxia exposure or eRNA treatment but is reduced upon pre-treatment with RNaseA, 2 hrs before hypoxia exposure. Furthermore, immunohistochemistry/or immunofluorescence studies were also performed to identify the presence of CD31 (PECAM-1), Neutrophil Elastase (NE) (Fig 5D and 5E), CD11/CD18 (MAC-1) and CD41 (megakaryocyte and platelet) markers on myocardial tissues (Fig 5E and 5F). The relative quantitative densitometry analysis of the respective images (Fig 5A–5F) is presented in S1A–S1F Fig.

It was observed that the increased leukocyte infiltration (depicted by PECAM-1, NE and CD11/CD18) and platelet accumulation (CD41) due to hypoxic stress was ameliorated by RNaseA pre-treatment, whereas eRNA was used as the positive control.

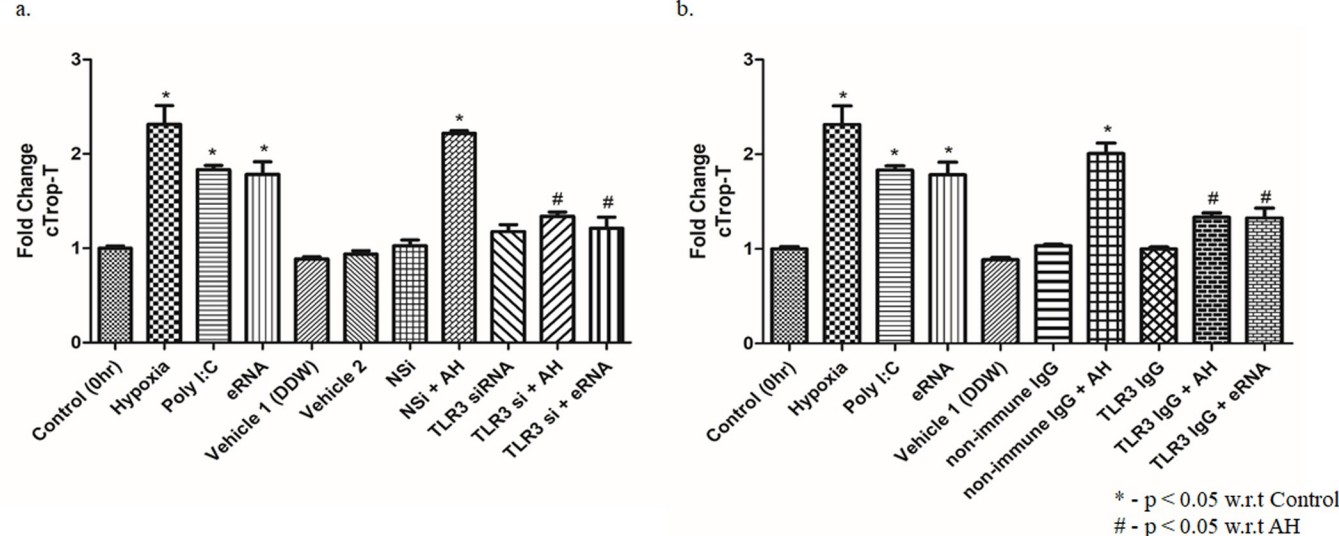

**Fig 4. AH induced eRNA facilitates release of cTrop-T via TLR3 signalling.** ELISA estimation of circulating cTrop-T in plasma was done after (a) TLR3 IgG treatment (40 mg/kg BW; 2 hrs prior to AH exposure). Poly I:C (1 mg/kg BW; 2 hrs prior to AH exposure) and eRNA (80 μg/kg BW; 6 hrs prior to AH exposure) treatments were used as positive controls and (b) TLR3 siRNA treatment. Non-specific (NS) siRNA control and non-immune IgG control were used similarly in another group of mice. Data are shown as mean ± SEM (n = 5/group/treatment) from one experiment representative of three independent experiments, all performed in triplicate. One-way ANOVA revealed statistical significance in the results ($^*p < 0.05$ groups w.r.t control; $^\#p < 0.05$ groups w.r.t Hypoxia and eRNA).

## 6. AH induced cTrop-T release is mediated by activation of caspase-3

Traditionally, MI was shown as manifestation of cardiomyocyte necrosis [26,27]. However, the concept of caspase-dependent regulated necrosis (CDRN) involving liberation of

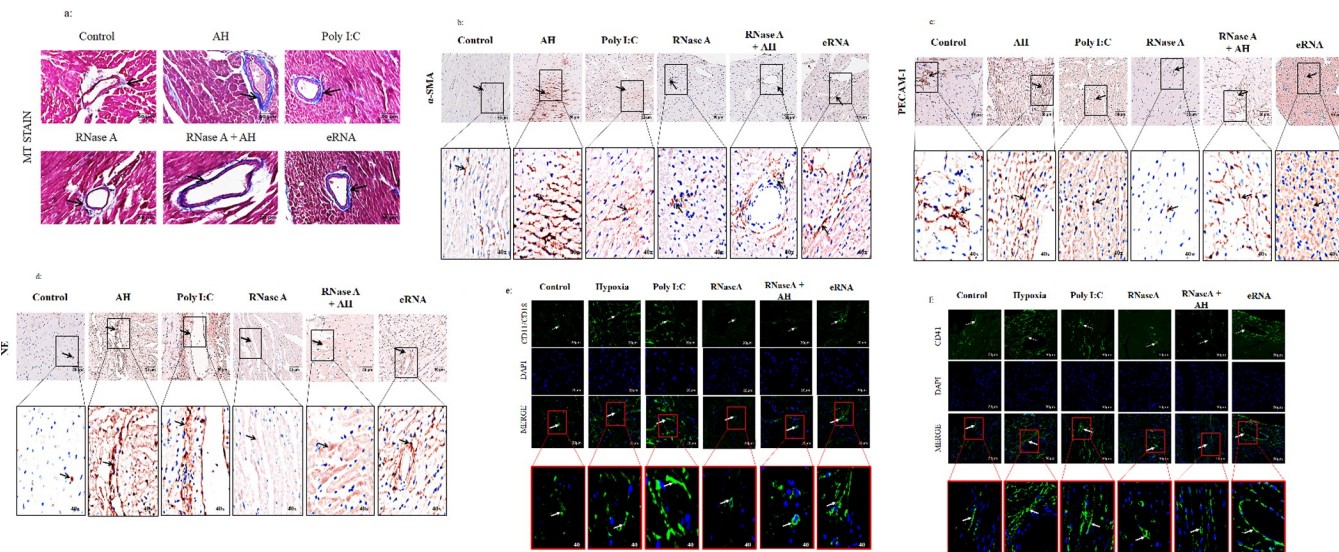

**Fig 5. Hypoxia induced eRNA stimulates collagen accumulation, leucocyte infiltration and activation in cardiomyocytes.** (a) MT staining (Collagen stained purple whereas tissue was stained pink); immunohistochemistry/immunofluorescence analysis of (b) α-SMA, (c) PECAM-1, (d) NE, (e) CD11/CD18 and (f) CD41 in myocardial tissue sections have been presented with higher magnification images in the subset. Magnification of the images in subset is 40X. Images were acquired at 20X and 40X resolution. DAPI (blue) was used as a nuclear stain. Data are representative of three independent times with three mice per experiment. (Scale bar: 50 μm).

nucleosomes and attached DAMPs with no fragmentation of nuclei, etc [28] correlates with myocardial injury observed upon AH exposure. Hence, the activation and release of caspase-3 was studied in this context. Activated Caspase-3 was analysed in plasma and heart tissue lysate of mice after AH exposure and other pre-treatments such as eRNA, RNaseA, TLR3 siRNA and TLR3 IgG. eRNA dissolved in DDW (vehicle 1) and siRNA dissolved in RNALancer II solution (vehicle 2). Non-specific siRNA and non-immune IgG were used as negative controls and Poly I:C as positive control in the experiment. Our results elucidate that AH exposure and/or eRNA induction significantly activated Caspase-3 (Fig 6A and 6D; p<0.05). However, to validate the connection of TLR3 in caspase-3 release, mice were treated with TLR3-immunoneutralizing antibody two hrs before AH exposure and the caspase-3 was quantified in plasma and tissue (heart) lysate by ELISA. Our result illustrated that the pre-treatment with TLR3-immunoneutralizing antibody significantly inhibited caspase-3 release (Fig 6B and 6E; p<0.05). To affirm the connection of TLR3 we performed *Tlr3* gene silencing *in vivo* by using *Tlr3* siRNA, and the plasma and tissue lysate level of caspase-3 was estimated. Our results showed that AH-induced caspase-3 level was substantially reduced in *Tlr3*-silenced mice as compared to the control (Fig 6C and 6F; p<0.05). This indicates that AH induced release of activated caspase-3 is mediated through TLR3.

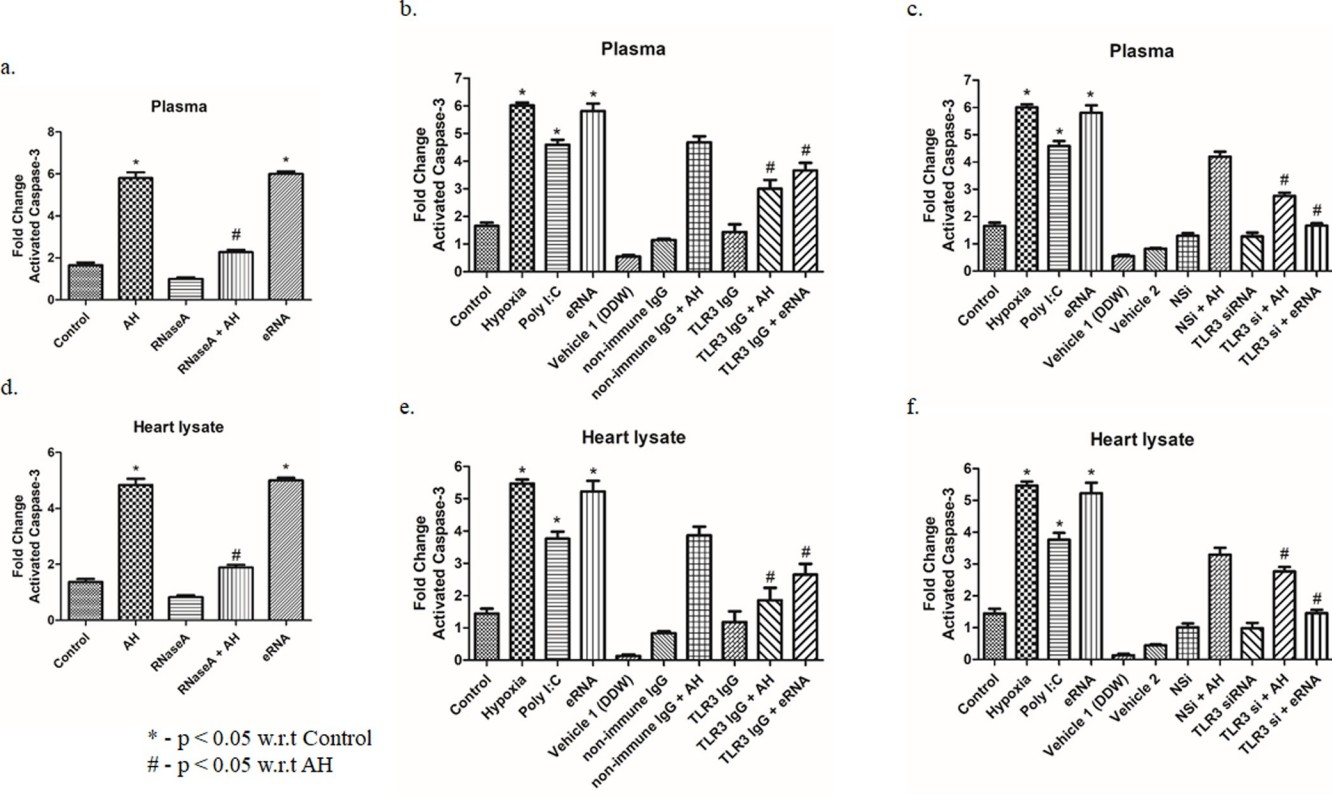

**Fig 6. AH induced cTrop-T release is mediated by activation of caspase-3.** Estimation of activated Caspase-3 was done after (a) eRNA injection (80 µg/kg BW; 6 hrs prior to AH exposure) and RNaseA pre-treatment (1mg/kg BW; 2 hrs prior to AH exposure) in plasma, (b) TLR3 IgG treatment in plasma (c) TLR3 siRNA treatment (40 mg/kg BW; 2 hrs prior to AH exposure) in plasma, (d) eRNA injection (80 µg/kg BW; 6 hrs prior to AH exposure) and RNaseA pre-treatment (1mg/kg BW; 2 hrs prior to AH exposure) in heart lysate, (e) TLR3 IgG treatment (40 mg/kg BW; 2 hrs prior to AH exposure) in heart lysate and (f) TLR3 siRNA treatment in heart lysate. Data are shown as mean ± SEM (n = 5/group/treatment) from one experiment representative of three independent experiments, all performed in triplicate. One-way ANOVA revealed statistical significance in the results (*p< 0.05 groups w.r.t control; #p<0.05 groups w.r.t AH).

## Discussion

This study explored the mechanism of myocardial injury due to exposure of AH. The precise role of eRNA and its receptor TLR3 has been evaluated in the induction of cardiomyocyte necrosis as well as infiltration of leukocytes into the cardiomyocytes that lead to tissue damage and release of cTrop-T. The functional implications of TLR3 signaling was studied using genetic as well as pharmacological approaches. To investigate the importance of eRNA, we injected RNaseA *in vivo*, which significantly diminished not only AH-induced release of SI and cardiac injury markers but also collagen accumulation, leukocyte infiltration and activation in cardiomyocytes. Our genetic study, in vivo silencing of TLR3 also abrogated the AH-induced release of cTrop-T due to myocardial injury. Thus, eRNA-TLR3 signalling pathway could be the potential target for the amelioration of AH-induced AMI.

The inflammatory response triggered by myocardial injury, primarily for healing, may initiate cardiac dysfunction [29] by releasing DAMPs from injured myocardia [30] which triggers innate immune response involving amplified cytokine expression, neutrophil infiltration [31] and cardiomyocyte apoptosis/necrosis [32]. Here, we have observed that AH exposure facilitates release of sterile inflammatory molecules such as eRNA, eDNA, HMGB1, HSPs, vWF, and s100b into the circulation. TLR3, a known receptor for ds-viral RNA, which recognizes endogenous RNA was released from necrotic cells in vitro [33]. This cellular RNA, released from injured cardiomyocytes or ischemic myocardium induces cytokine production [34]. Here, we have shown that eRNA released from the injured murine myocardial tissue significantly escalates cTrop-T release *via* TLR3 signalling. Whereas, RNaseA pre-treatment or TLR3 silencing significantly abrogated cTrop-T release in the circulation.

cTrop-T and myoglobin are specific cardiac markers used for diagnosis of myocardial injury. In contrast to other cell types, cardiomyocytes are more prone to necrosis triggered by calcium or oxygen insult that result in the release of cTrop-T along with other cellular components [35]. Here, we observed cTrop-T elevation upon AH exposure as early as six hrs suggesting myocardial injury is an early event of AMI.

Stress mediated cardiomyopathy involves agranulocytes infiltration, microvascular dysfunction, and cytokine storm which consequently trigger a strong immune response. The process of leukocyte activation due to inflammation following the recruitment of neutrophil is mediated by selectins, a member of β2-intergin family (MAC-1), which regulates neutrophil and leukocyte adhesion via ICAM-1 on endothelial cells [36]. Neutrophil elastase (NE) is expressed not only by neutrophils but also by monocytes, macrophages and endothelial cells [37] and this modulates cytokine activity during inflammation. Its elevated expression has also been reported in CVDs [38], atherosclerotic plaques [37] and other cardiac complications. Neutrophils serve as host defenses against viral pathogens and are activated by poly I:C to mediate immune response [39], promotes megakaryocyte fragmentation into platelets, leading to platelets activation and subsequent thrombosis in coronary arteries [40]. A disarray in the coagulation and the fibrinolytic system due to myocardial injury has been observed as increased D-dimer (thrombogenesis indicator), fibrinogen (coagulation factor), vWF (endothelial dysfunction marker) and CD41 level (platelet and megakaryocyte surface marker) [41]. PECAM-1, expressed on leukocytes, macrophages and some T-cells is implicated in their infiltration and implicated in the rat model of myocardial [42] and intestinal I/R injury [43]. Here, we observed marked leukocytes infiltration in cardiomyocytes due to AH exposure. However, the pre-treatment of RNaseA two hrs before of AH exposure mitigates it. This shows the role of eRNA in leukocytes infiltration in cardiomyocytes during AH exposure.

Cytokine induced conversion of fibroblasts to myofibroblasts initiates the process of collagen production [44] as well as its accumulation which results in myocardial stiffness and

arteriosclerosis. This also dampens the Windkessel effect, which causes a rise in the pulse as well as systolic pressure and subsequently upsurge the risk of MI, stroke and other CVDs. Our results showed marked collagen accumulation in coronary vasculature due to AH exposure or/ eRNA induction. However, RNaseA pre-treatment (two hrs before exposure to AH) significantly inhibits it. This demonstrates the role of eRNA in collagen accumulation as well as stiffness of coronary vasculature due to AH exposure.

Caspase-3, a predominant apoptotic marker, is involved in CDRN which induces the release of nucleosomes and DAMPs. These remains in the inactive form in cytosol and becomes active by self-sustained proteolytic autocatalysis [45]. Here, we observed the presence of active caspase-3 in heart tissue due to AH exposure as well as eRNA induction which was significantly ameliorated by either RNaseA treatment and or/ TLR3 silencing.

Briefly, we delineated that AH-induced SI molecules elevations specially eRNA leads to neutrophil activation and leucocyte infiltration in myocardial tissue, which causes cell injury and death via the caspase dependent pathways. The resulting myocardial necrosis releases cTrop-T from the sarcomeres into the circulation where by the tropomyosin remains attached to actin filaments and prevents the binding of myosin; hence developing contractile dysfunction. The subsequent collagen accumulation culminates into myocardial stiffness and dysfunction which ultimately may lead to heart failure. Here, we have validated that the release of eRNA under AH condition triggers TLR3 which facilitate myocardial injury as well as necrosis through leucocyte infiltration. We further showed that this process could mitigate though RNaseA pre-treatment and/ or TLR3 silencing. Thus, this raises a note-worthy possibility of RNaseA or TLR3-silencing therapy as a potential therapeutic option in the prevention of AH-induced AMI or CVDs.

## Supporting information

**S1 Fig.** Densitometry analysis of (a) MT staining depicting collagen, (b) immunohistochemistry for α-SMA, (c) immunohistochemistry for PECAM, (d) immunohistochemistry for NE, (e) immunofluorescence for CD11/CD18 and (f) immunofluorescence for CD41. Data are shown as representative of three independent experiments, all performed in triplicate. One-way ANOVA revealed statistical significance in the results (*$p < 0.05$ groups w.r.t control; #$p < 0.05$ groups w.r.t AH).
(PDF)

**S2 Fig. Agarose gel image depicting TLR3 expression upon TLR3 siRNA knockdown.** Lane 1- Control (no siRNA treatment), Lane 2- TLR3 siRNA treatment and Lane 3- non-specific siRNA treatment.
(PDF)

## Acknowledgments

We would like to acknowledge and thank various division of DIPAS-DRDO who helped us in the work by providing permission to use certain instruments in their labs. I thank Maneeza Khan for helping in manuscript editing.

**Disclosure**

The abstract of this paper was presented at the FIPSPHYSIOCON 2017 Conference as an oral presentation with pilot study findings. The abstract was published in conference proceedings.

## Author Contributions

**Conceptualization:** Gausal A. Khan.

**Data curation:** Saumya Bhagat.

**Formal analysis:** Gausal A. Khan.

**Funding acquisition:** Gausal A. Khan.

**Investigation:** Saumya Bhagat, Indranil Biswas, Gausal A. Khan.

**Methodology:** Gausal A. Khan.

**Project administration:** Gausal A. Khan.

**Resources:** Gausal A. Khan.

**Supervision:** Gausal A. Khan.

**Validation:** Md Iqbal Alam, Gausal A. Khan.

**Writing – original draft:** Saumya Bhagat.

**Writing – review & editing:** Madiha Khan, Gausal A. Khan.

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
