## [Decision Letter · Decision Letter 0]

18 Jun 2021

PONE-D-21-14652

Key role of Extracellular RNA in hypoxic stress induced myocardial injury

PLOS ONE

Dear Dr. Khan,

Thank you for submitting your manuscript to PLOS ONE. After careful consideration, we feel that it has merit but does not fully meet PLOS ONE’s publication criteria as it currently stands. Therefore, we invite you to submit a revised version of the manuscript that addresses the points raised during the review process.

We look forward to receiving your revised manuscript.

Kind regards,

Aftab Ahmad, Ph.D.

Academic Editor

PLOS ONE

2. In your Methods section, please provide methods of animal sacrifice.

Reviewers' comments:

Reviewer's Responses to Questions

**Comments to the Author**

1. Is the manuscript technically sound, and do the data support the conclusions?

Reviewer #1: Yes

Reviewer #2: Partly

Reviewer #3: No

2. Has the statistical analysis been performed appropriately and rigorously? 

Reviewer #1: Yes

Reviewer #2: Yes

Reviewer #3: Yes

3. Have the authors made all data underlying the findings in their manuscript fully available?

Reviewer #1: Yes

Reviewer #2: No

Reviewer #3: No

4. Is the manuscript presented in an intelligible fashion and written in standard English?

Reviewer #1: Yes

Reviewer #2: No

Reviewer #3: No

5. Review Comments to the Author

Reviewer #1: Review Comments to the Author

Authors have studied the key role for eRNA and TLR3 in hypoxia-induced myocardial ischemic injury. The authors presented the mechanism of cTrop-T release and progression of acute myocardial infarction through TLR3-caspase -3 pathway in a murine model of acute hypoxia. The proposed aim and choice of testing the eNA is novel and has current interest in the field of cardiovascular biology. Although there is data provided in support of the aim and objectives, I have some concerns with the manuscript, primarily with methodology and some aspects where statements appear to confuse cause and effect. The paper will be of significant interest, provided the issues below are addressed and modifications are made.

Major comments:

1. The background should clearly state the purpose of the study.

2. In the abstract section, (page 2, line 25) the sentence, “Although the diagnosis is based on……” is incomplete.

3. Details of methodology are missing in several places.

a. How long were these animals exposed to acute hypoxia conditions?

b. In the method section, authors mentioned that 80ug of eRNA and eDNA were injected into the animal by intravenously through the tail vein. What is the starting material for isolating eRNA and eDNA from plasma? What is the yield eRNA and eDNA isolated from plasma of exposed animal?

c. Is eDNA and eRNA from each animal or is it from pooled samples of exposed animals? Please give addition details in methods section.

d. The methodology for isolating eDNA reported in the manuscript is missing. Please add this to methodology.

e. In addition, the time point for the proposed experiments are not described. Please provide specific time point for histological and immunochemical study. Is the time point the same for all parameters?

f. Under the methodology section, line 142, ELISA, the authors should provide detailed information about specific antibodies (primary and secondary antibody) and the dilution used for the assay. Methods do not have proper references and are not explained in detail.

g. For Immunofluorescence, please detail antibody information for both primary and secondary antibodies for CD31, CD11/CD18, neutrophil elastase etc.

h. Please provide higher magnification image showing leukocyte infiltration along with low magnification images.

Minor comments

4. In page 7, line 143 50ug plasma, it should be in uL

5. The quality of the Figure 1-4 is poor. Please maintain the same consistency in numbering all figure panels.

6. The manuscript needs critical editing for language and syntax errors.

Results from the present study agree with previously published reports. The above issues do not abrogate the potential importance of the paper; however, I do think it would be very interesting to see the modified manuscript including the answers for these queries.

Reviewer #2: This manuscript by Bhagat S et al., titled “key role of extracellular RNA... myocardial injury” investigates the role of extracellular RNA (eRNA) in causing myocardial injury caused by exposure of mice to acute hypoxia (AH). The authors show that AH results in increase in eRNA leading to increased cardiac troponin-T (cTrop-T) in plasma through a TLR3-dependent pathway and that blocking the signal through either RNase treatment, or TLR-3 neutralizing antibodies or siRNA against TLR3 reduces levels of cTrop-T. They also show that poly(I:C) a surrogate of eRNA causes a similar increase in cTrop-T. While the studies are interesting the manuscript is difficult to follow and is poorly written. Experimental details are lacking in several places.

Comments:

1. Animals were exposed to acute hypoxia (~8.5% O2) simulating an altitude of 7628 m. The relevance of this level of hypoxia is not clear. Please add details of hypoxic exposure in the Methods section.

2. In Figure 1, fold change in eRNA and eDNA have been shown. Please give the actual amount/unit of the plasma. Assuming that standards were run in ELISA as per reference, the amount of HMGB1, vWF, HSP70, HSP90 and s100b should be given. Details of the methodology are sketchy. For instance, it is stated that 50 ug of plasma or tissue lysate was used in ELISA. What is the volume of plasma used and whether it was diluted to achieve the desired protein concentration? Also, ‘expression analysis’ of SI markers are mentioned. These are eRNA/eDNA, which are not necessarily expressed genes. Most measured molecules are protein, levels of which change following AH. The title of the legend and the legend needs to be changed accordingly.

3. Figure 3 shows changes in troponin. Please correct the y-axis to specify the type of troponin and provide quantitation. The time point of evaluation should also be given in the figure legend. Additionally, it is not clear how much eRNA, eDNA or Poly(I:C) was used as a positive control. Please add these details to the figure legend as well.

4. Figure 4: while the figure shows expression of troponin-T, apparently protein levels were determined by ELISA. Please label accordingly with quantitation rather than fold change. Please also provide details in the figure legends on when the TLR3 neutralizing antibodies were added, and the duration of hypoxic exposures used.

5. Figure 5 (I) shows PECAM -1 and neutrophil elastase for staining of endothelial cells and neutrophils respectively. Apparently PECAM-1 should be in the blood vessels. The images are not of sufficient resolution to appreciate the staining differences in the different groups. Please also provide high resolution images to demonstrate specificity in staining. Similarly, also provide high resolution images of the NE stained sections so the reader can appreciate specificity.

6. Figure 5 (II) shows staining of CD11/CD18 and CD41. The structure within these images is not discernable. Please provide high resolution images to show structures.

7. It is stated that TLR3 levels were knocked out in vivo using siRNA. While siRNA approaches are known to knockdown genes of interest, they are less likely to knockout genes. Please provide more details and demonstrate efficient knockdown or knockout of TLR3.

8. Figure 6 shows fold changes in activated caspase-3 in plasma and heart tissue lysates. Please plot the plasma and tissue lysate findings separately. Please provide details of the commercial kit used. If an ELISA based kit was used, provide quantitation.

9. In writing figure legends, avoid describing the results.

Minor comments:

Sentences are incomplete in several places and the language has syntax errors.

In the ‘Abstract’ line 25 appears incomplete.

Throughout the manuscript PECAM-1 is written as PECAM. Please correct it.

Page 5, lines 107-119. Please write complete sentences.

Page 9, lines 181, please correct the word ‘compression’

Page 9, lines 191, please correct the word ‘dependant’.

Page 10, lines 214, inappropriate use of the word ‘whereas’.

Page 11, lines 241, consider using an alternate word of ‘abridged’. The sentence, as written, lacks clarity.

Page 12, lines 251, please clarify the word ‘caNerspase-3’ or change the title to make it understandable.

Page 12, lines 258-259, please complete the sentence or correct the syntax.

Reviewer #3: This study aims to evaluate the effects of acute hypoxia on cardiomyocytes and role of eRNA in this process through potential activation of TLR3-Caspase 3. The authors try to link this in vivo hypoxia model to clinical acute myocardial pathology in human. This by itself dampen the enthusiasm for the initial question and the readouts collected by the authors. It would have been more logical if authors would have initiated the question using this animal model as to determine the effect of hypoxia on cardiac microvascular inflammation and the role of eRNA-TLR3 axis in this process without the necessity to discuss troponin and cardiomyocytes necrosis etc which was never confirmed based on the data shown.

Other major criticism that might help the impact of this study are as below:

1- All experimental readouts are based on ELIZA and Calorimetric assays. It is necessary for authors to include and add more supporting data even as supplemental data including but not limited to mRNA/protein analysis using PCR and western analysis.

2- There is no data included to confirm some of the proposed methodological approaches have been effective. This will help to better judge some of the data presented in Figures 2, 3 and 4.

3- In Figure 3a, why troponin is lower in RNAse treated control animals when compared to sham and with no hypoxia.

4- In Figure 3b, why the reduction in troponin in animals treated with RNAse and DNAse seems to be similar?

5- Figures 3 legend mentions that eRNA is involved in hypoxia induced release of cTrop-T while on figure it mentions “expression”.

6- Not clear what does “Cardiomyocytes activation” means in figure 5?

6. PLOS authors have the option to publish the peer review history of their article (what does this mean?). If published, this will include your full peer review and any attached files.

Reviewer #1: **Yes: **NITHYA MARIAPPAN

Reviewer #2: No

Reviewer #3: No

---

## [Author Response · Author response to Decision Letter 0]

4 Aug 2021

First Revision – author’s’ response 2rd August, 2021

All editing in the manuscript is presented as track change.

I am indebted to you for the valuable comments for overall improvement of the manuscript. Accordingly, in this revised paper, I give due weightage to comments and the resultant responses are:

The answer to the Reviewer # I

Major Comments:

Point: 1

The background should clearly state the purpose of the study.

Author Response: I wish to thank reviewer for the comments. The background of the study is mentioned in abstract (page 1 line 25-28) and in Introduction (page 4 line 69-93).

Point: 2

In the abstract section, (page 2, line 25) the sentence, “Although the diagnosis is based on……” is incomplete.

Author Response: I thank the reviewer for the pointing out the mistake. We have addressed the concern and necessary correction was made in the abstract (page 2, line 27-28).

Point: 3

Details of methodology are missing in several places.

a. How long were these animals exposed to acute hypoxia conditions?

b. In the method section, authors mentioned that 80ug of eRNA and eDNA were injected into the animal by intravenously through the tail vein. What is the starting material for isolating eRNA and eDNA from plasma? What is the yield eRNA and eDNA isolated from plasma of exposed animal? 

c. Is eDNA and eRNA from each animal or is it from pooled samples of exposed animals? Please give addition details in methods section.

d. The methodology for isolating eDNA reported in the manuscript is missing. Please add this to methodology.

e. In addition, the time point for the proposed experiments are not described. Please provide specific time point for histological and immunochemical study. Is the time point the same for all parameters?

f. Under the methodology section, line 142, ELISA, the authors should provide detailed information about specific antibodies (primary and secondary antibody) and the dilution used for the assay. Methods do not have proper references and are not explained in detail. 

g. For Immunofluorescence, please detail antibody information for both primary and secondary antibodies for CD31, CD11/CD18, neutrophil elastase etc.

h. Please provide higher magnification image showing leukocyte infiltration along with low magnification images.

Author Response: I thank reviewer for raising these concerns. The details as per the comments of the reviewer have been incorporated in the methodology section of the manuscript as follows:

a. In time dependent study, these animals were exposed to AH for 0, 6, 12, 24 hrs., 3 and 7 days which are mentioned in “Acute hypoxia and other treatments” (page 6, line 127-133). However, we found that as early 6 hrs. of AH exposure significantly up regulated all the protein molecules. For our further experiments, we chose to use the 6 hrs. time point of hypoxia exposure for mice model in all the experiments unless otherwise mentioned as on Page 10, line 207-208 which is in line of our previous publication reference [Bhagat et al, BCMD,84 (2020),102459]. 

b. In brief, approximately 1mL of blood we had collected from each AH exposed mice where we were able to separate approximately 0.55 mL of plasma. Therefore, 0.55mL/mice plasma was our starting materials for isolation of eRNA and eDNA. The yield of eRNA and eDNA in AH treated each mouse was approximately 100 µg/mL and 60 µg/mL respectively. Therefore, yield of eRNA & eDNA was 55 µg/mice & 33 µg/mice respectively. The injected doses of eRNA or eDNA were 80 µg/kg BW in our experiments. Each mice weighted approximately 20g. Therefore, the required amount of eRNA or eRNA were 1.6 µg/mice which we obtained form once mice. We had performed similar experiments in our previous manuscript [Bhagat et al, BCMD, 84 (2020),102459]. We hope that we have clearly explain to the reviewer. 

c. The eRNA and eDNA used in our experiments were isolated form each AH exposed animal then pooled and used. The same has been mentioned in the methods section (page 7, line 143-146). Similar method, we used in our previous experimentation [Bhagat et al, BCMD, 84 (2020), 102459].

d. In our experiments, the eDNA was isolated though a commercially available kit (Quick-cfDNA, Zymo Research) as per the manufacture instruction and it is also mentioned in the manuscript (page 7, line 153-156). Same method we used in our previous experimentation also [Bhagat et al, BCMD, 84 (2020), 102459].

e. Animals were exposed to hypoxia for 0, 6, 12 and 24 hrs., 3 days and 7 days. However, we used 6 hrs. time points in all the experiments as well as immune histochemistry and IF study because as early in this time point all the parameters are significantly high.

f. Details of specific primary and secondary antibodies used along with their dilutions have been mentioned in the methodology section (page 8, line 161-164).

g. Details of specific primary and secondary antibodies used for IHC and IF along with their dilutions have been mentioned in the methodology section (page 8, line 182-183).

h. Higher resolution image (40X) is added as sublet in respective picture.

Point: 4

In page 7, line 143 50ug plasma, it should in uL

Author response: I wish to thank the reviewer for pointing out the mistake. It is now corrected in page 7 line 158 (50 µg total protein per well in equal volume). 

Point: 5

The quality of the Figure 1-4 is poor. Please maintain the same consistency in numbering all figure panels.

Author response: I wish to thank the reviewer for pointing out the mistake. The figures 1-4 have been modified as per the reviewers’ suggestions, same numbering and formatting has been maintained in all the figures. 

Point: 6

The manuscript needs critical editing for language and syntax errors.

Author response: I thank the reviewer for the concern. The manuscript has been proofread and syntax errors taken care.

The answer to the Reviewer # 2

Comments

Point: 1

Animals were exposed to acute hypoxia (~8.5% O2) simulating an altitude of 7628 m. The relevance of this level of hypoxia is not clear. Please add details of hypoxic exposure in the Methods section.

Author Response: I thank the reviewer for raising the concern. In brief, animals were subjected to AH in a specially fabricated animal decompression chamber in which conditions were equivalent to atmospheric conditions at an altitude of 7628 m (282 mm Hg), with an O2 content of ~8.5%, as previously described [Biswas et al, Eur. J. Immunol. 2015. 45: 3158–3173]. The temperature and humidity were maintained at 25 ± 3°C and 55% ± 5%, respectively. In our study, we explored the effect of AH in MI. Therefore, we used same AH conditions which have been standardized in our lab and previously published in our papers [Bhagat et al, BCMD, 84 (2020), 102459 & Biswas et al, Eur. J. Immunol. 2015. 45: 3158–3173]. It is also mentioned in the manuscript page 6 line 127- 130). 

Point: 2

In Figure 1, fold change in eRNA and eDNA have been shown. Please give the actual amount/unit of the plasma. Assuming that standards were run in ELISA as per reference, the amount of HMGB1, vWF, HSP70, HSP90 and s100b should be given. Details of the methodology are sketchy. For instance, it is stated that 50 ug of plasma or tissue lysate was used in ELISA. What is the volume of plasma used and whether it was diluted to achieve the desired protein concentration? Also, ‘expression analyses of SI markers are mentioned. These are eRNA/eDNA, which are not necessarily expressed genes. Most measured molecules are protein, levels of which change following AH. The title of the legend and the legend needs to be changed accordingly.

Author Response: I thank the reviewer for raising the concern. The absolute values of eRNA and eDNA at different time-points of hypoxia exposure are mentioned below:

Time-point (hr.) eRNA (µg/ml) Time-point (hr.) eDNA (µg/ml)

0 0.87 ± 0.09 0 0.28 ± 0.01

6 2.07 ± 0.37 6 0.83 ± 0.03

12 1.73 ± 0.12 12 1.22 ± 0.05

24 3.14 ± 0.51 24 1.24 ± 0.05

It is further mentioned that we assayed the different protein in plasma or tissue lysate by ELISA with standard established procedures we used in our several previous publications papers [Bhagat et al, BCMD, 84 (2020), 102459 & Biswas et al, Eur. J. Immunol. 2015. 45: 3158–3173]. We did quantitative ELISA with standard, but the data were presented in fold of expression. This is not a clinical study we would rather a basic science research where fold change is signifies the results. We had done similar study and results also presented previously. [Bhagat et al, BCMD, 84 (2020), 102459; Biswas et al, Eur. J. Immunol. 2015. 45: 3158–3173; Biswas et al, BCMD, 49 (2012) 92–10; Singh et al, Biochemistry 2014, 53, 115−126]. As per the reviewer suggestion all the corrections are incorporated. It is now corrected in page 7 line 158 (50 µg total protein per well in equal volume). Figure-1 shows the relative expression of different SI markers, nucleic acids and proteins, upon exposure to different durations of hypoxia after statistical analysis to obtain significance. The term “expression analysis” has been corrected in the figure legend no.1 (page 18, line 390-391). I hope the reviewer will consider it.

Point: 3

Figure 3 shows changes in troponin. Please correct the y-axis to specify the type of troponin and provide quantitation. The time point of evaluation should also be given in the figure legend. Additionally, it is not clear how much eRNA, eDNA or Poly(I:C) was used as a positive control. Please add these details to the figure legend as well.

Author response: I wish to thank the reviewer for the suggestions. As per the suggestions, the figure-3 have been edited. We have shown the expression of cardiac Troponin-T (cTrop-T) in the plasma of mice after different treatments. The dosage concentration for each treatment has been mentioned in the figure legend (page 18, line 409-412). AH exposure in every experiment is 6 hrs. 

Point: 4

Figure 4: while the figure shows expression of troponin-T, apparently protein levels were determined by ELISA. Please label accordingly with quantitation rather than fold change. Please also provide details in the figure legends on when the TLR3 neutralizing antibodies were added, and the duration of hypoxic exposures used.

Author response: I thank the reviewer for the suggestion. The graphs of figure-4 have been edited as per reviewer suggestion. We have shown the expression of cardiac Troponin-T (cTrop-T) in the plasma of mice after TLR3 siRNA and TLR3 IgG treatments.

It is further mentioned that we assayed the Troponin-T by ELISA with standard established procedures we used in our several previous publications papers [Bhagat et al, BCMD, 84 (2020), 102459 & Biswas et al, Eur. J. Immunol. 2015. 45: 3158–3173]. We did quantitative ELISA, but the data were presented in fold of expression. This is not a clinical study we would rather a basic science research where fold change is signifies the results. We had done similar study and results also presented previously. [Bhagat et al, BCMD, 84 (2020), 102459; Biswas et al, Eur. J. Immunol. 2015. 45: 3158–3173; Biswas et al, BCMD, 49 (2012) 92–10; Singh et al, Biochemistry 2014, 53, 115−126]. As per the reviewer suggestion all the corrections are incorporated Details of TLR3 neutralizing antibody treatment has been included in the figure legend (page 18, line 423-424). 

Point: 5

Figure 5 (I) shows PECAM -1 and neutrophil elastase for staining of endothelial cells and neutrophils respectively. Apparently PECAM-1 should be in the blood vessels. The images are not of sufficient resolution to appreciate the staining differences in the different groups. Please also provide high resolution images to demonstrate specificity in staining. Similarly, also provide high resolution images of the NE-stained sections so the reader can appreciate specificity.

Author response: I thank the reviewer for the concern. As per suggestion high resolution image also included in the figure as subset (Supplementary Figure 1 a-c). 

Point: 6

Figure 5 (II) shows staining of CD11/CD18 and CD41. The structure within these images is not discernable. Please provide high resolution images to show structures.

Author response: I thank the reviewer for the concern. As per suggestion high resolution image also included in the figure as subset (Supplementary Figure 1 d, e).

Point: 7

It is stated that TLR3 levels were knocked out in vivo using siRNA. While siRNA approaches are known to knockdown genes of interest, they are less likely to knockout genes. Please provide more details and demonstrate efficient knockdown or knockout of TLR3.

Author response: I thank the reviewer for raising the concern. The details results are below. We used the similar methodology in our previously polished papers and details are also there [Bhagat et al, BCMD, 84 (2020), 102459; Biswas et al, Eur. J. Immunol. 2015. 45: 3158–3173]. For this reason, we did not include TLR3 siRNA mediated knockout data. Appropriate changes in the manuscript have been made. Page 8 line 168-174 & Page 11 line 240-249. Upon TLR3 siRNA treatment we observed significant silencing of the TLR3 gene, and proceeded with further experiments.

This agarose gel image shows the expression of TLR3 gene under different conditions; Lane 1- Control (no siRNA treatment), Lane 2- TLR3 siRNA treatment and Lane 3- non-specific siRNA treatment.

Point: 8

Figure 6 shows fold changes in activated caspase-3 in plasma and heart tissue lysates. Please plot the plasma and tissue lysate findings separately. Please provide details of the commercial kit used. If an ELISA based kit was used, provide quantitation.

Author response: I thank the reviewer for raising the concern.

Graphs of Figure-6 have been modified as per the reviewer suggestions and changes made to the manuscript accordingly. Estimation was done using the Caspase-3 Fluorometric Assay Kit, BioVision Inc. (Page 9 line 196-198). Since the samples have been collected after various treatments and hypoxia exposure to mice, control mice with no treatment or hypoxia exposure were taken as assay control where caspase-3 activity was un induced. Fold change analysis of all sample sets was done in comparison to control.

Point: 9

In writing figure legends, avoid describing the results.

Author response: I thank the reviewer for raising the concern. Figure legends have been modified as per the suggestions of the reviewer (Page 18-20).

Point: 10

Sentences are incomplete in several places and the language has syntax errors.

In the ‘Abstract’ line 25 appears incomplete.

Throughout the manuscript PECAM-1 is written as PECAM. Please correct it.

Page 5, lines 107-119. Please write complete sentences.

Page 9, lines 181, please correct the word ‘compression’

Page 9, lines 191, please correct the word ‘dependant’.

Page 10, lines 214, inappropriate use of the word ‘whereas’.

Page 11, lines 241, consider using an alternate word of ‘abridged’. The sentence, as written, lacks clarity.

Page 12, lines 251, please clarify the word ‘caNerspase-3’ or change the title to make it understandable.

Page 12, lines 258-259, please complete the sentence or correct the syntax.

Author response: I thank the reviewer for raising the concern. All minor comments raised by the reviewer have hereby been addressed in the manuscript.

Abstract line 27 has been completed correctly.

Page 5, lines 105-125, appropriate changes as per reviewer comments have been incorporated.

Page 9, line 202, the word “compression” has been corrected to “comparison”.

Page 10, line 213, the word “dependant” has been corrected.

Page 11, line 235, the sentence has been rectified. 

Page 12, line 262, the word abridged has been replaced and the sentence modified accordingly.

Page 12, line 273, the word has been corrected to Caspase-3.

Page 13, lines 279-282, sentence has been corrected for errors.

The answer to the Reviewer # 3

Comments

Point: 1

All experimental readouts are based on ELIZA and Calorimetric assays. It is necessary for authors to include and add more supporting data even as supplemental data including but not limited to mRNA/protein analysis using PCR and western analysis.

Author response: I thank the reviewer for the comments. ELISA is a state of art technique that is why used it. In spite of it in support of the ELISA we used Immunohistochemistry as well as immune florescent studies. These are the direct proof of the expression of the different molecules. However, the protein detection/analysis by Western Blotting are not most reliable. Even if we done the western blotting the we need to reproof by direct evidences. I belive ELISA, Immunohistochemistry as well as IF are more accurate and specific in this context. I hope reviewer will understand.

Point: 2

There is no data included to confirm some of the proposed methodological approaches have been effective. This will help to better judge some of the data presented in Figures 2, 3 and 4.

Author response: I thank reviewer for the concern. In this manuscript we used Pharmacological approach (inhibitors) as well as genetic approach (siRNA mediated gene silencing) and the respective protein estimated by state-of-the-art method ELISA. We also sued immunohistochemistry and IF to revalidate the data. We have some limitation at this moment. I believe the methods were used are scientifically well established to proof the concept. I believe reviewer may please understand and consider. 

Point: 3

In Figure 3a, why troponin is lower in RNAse treated control animals when compared to sham and with no hypoxia.

Author response: I thank reviewer for the concern. We repeated these experiments several times and presented the data. Explanation: Basically, RNase A is an endoribonuclease that specifically degrades single-stranded RNA i.e., eRNA. In normal situation (control animals), a basal RNase A is present which maintain/regulate basal eRNA level which in turn control/maintain basal Troponin-T level in plasma. However, when we treated animals with exogenous RNase A (extra/excess than basal level), this might degrade basal eRNA. Therefore, basal eRNA level would decrease which in turn decrease the basal Troponin-T level than the normal or control animals. 

Point: 4

In Figure 3b, why the reduction in troponin in animals treated with RNAse and DNAse seems to be similar?

Author response: I thank the reviewer for the comments. We would like to explain our observation as follows, in figure 3a, there is reduction in circulating cTrop-T after RNaseA treatment as compared to control and after RNaseA pre-treatment prior to AH as compared to AH exposed animals. However, in figure 3b, we observe that DNase 1 treatment does not reduce circulating cTrop-T levels as compared to control nor after DNase 1 pre-treatment prior to AH as compared to AH exposed animals. Hence, the effect of RNaseA and DNase1 treatments are not similar. 

Point: 5

Figures 3 legend mentions that eRNA is involved in hypoxia induced release of cTrop-T while on figure it mentions “expression”.

Author response: I thank the reviewer for the comment. The discrepancy between the figures and figure legends have been rectified as per the suggestions of the reviewer.

Point: 6

Not clear what does “Cardiomyocytes activation” means in figure 5?

Author response: I thank the reviewer for the comment Addressing the comment of the reviewer, we would like to mention that in figure 5 we intend to show the infiltration and activation of leukocytes in the cardiomyocytes and not the activation of cardiomyocytes themselves.

---

## [Decision Letter · Decision Letter 1]

20 Sep 2021

PONE-D-21-14652R1Key role of Extracellular RNA in hypoxic stress induced myocardial injuryPLOS ONE

Dear Dr. Khan,

Thank you for submitting your manuscript to PLOS ONE. After careful consideration, we feel that it has merit but does not fully meet PLOS ONE’s publication criteria as it currently stands. Therefore, we invite you to submit a revised version of the manuscript that addresses the points raised during the review process.

As you can see from the comments, Reviewer 2 still has significant concerns, which have not been addressed. The reviewer’s request seems reasonable, and I encourage you to resubmit the manuscript after addressing each of the comments.

We look forward to receiving your revised manuscript.

Kind regards,

Aftab Ahmad, Ph.D.

Academic Editor

PLOS ONE

Journal Requirements:

Additional Editor Comments (if provided):

Reviewers' comments:

Reviewer's Responses to Questions

**Comments to the Author**

1. If the authors have adequately addressed your comments raised in a previous round of review and you feel that this manuscript is now acceptable for publication, you may indicate that here to bypass the “Comments to the Author” section, enter your conflict of interest statement in the “Confidential to Editor” section, and submit your "Accept" recommendation.

Reviewer #1: All comments have been addressed

Reviewer #2: (No Response)

Reviewer #3: All comments have been addressed

2. Is the manuscript technically sound, and do the data support the conclusions?

Reviewer #1: Yes

Reviewer #2: Partly

Reviewer #3: Yes

3. Has the statistical analysis been performed appropriately and rigorously? 

Reviewer #1: Yes

Reviewer #2: Yes

Reviewer #3: Yes

4. Have the authors made all data underlying the findings in their manuscript fully available?

Reviewer #1: Yes

Reviewer #2: No

Reviewer #3: Yes

5. Is the manuscript presented in an intelligible fashion and written in standard English?

Reviewer #1: Yes

Reviewer #2: Yes

Reviewer #3: Yes

6. Review Comments to the Author

Reviewer #1: (No Response)

Reviewer #2: This revised manuscript by Bhagat S et al., titled “key role of extracellular RNA... myocardial injury” investigates the role of extracellular RNA (eRNA) in causing myocardial injury caused by exposure of mice to acute hypoxia (AH). Changes have been made to the manuscript. However, there are still significant concerns that have not been addressed satisfactorily.

Comments:

1. In Figure 1, quantitation of the eDNA, eRNA as well as SI markers were requested. While the quantitation of eNA and eRNA were provided in the response letter, the plots were not changed to reflect quantitation. The authors fail to realize the significance of adding quantitation to the graphs in Figure 1. The intent is not to convince the reviewer but the reader. These numbers will serve as a guide to investigators in the field who may be doing similar experiments. The justification for not providing these numbers since it is a basic science study and not a clinical study is baseless and without any merit. In similar basic science studies, the authors have themselves published concentrations of eRNA and eDNA (Refs 1 and 13 of the manuscript). They have also published quantitative values for ELISA in these publications that they have authored. Interestingly, the y-axis values in Figure 1a and 1b correspond to the values provided in the response letter and are not ‘fold changes’ as indicated in the plots.

2. The supplementary figures 1a, 1b and 1c should be shown alongside Figure 5(I) to bring more clarity to the changes. The arrows in the lower and higher magnification images do not match and sometimes they show different regions (Supp Fig 1a: Arrows in Poly I:C and RNase A+AH pairs; Supp Fig 1b: Arrows in Poly I:C and RNase A+AH pairs; Supp Fig 1c: Arrows in Poly I:C and eRNA pairs). The figures in the main body of the manuscript should convey the reported findings without having to go to the supplementary data.

3. Similarly, the structures within Figure 5 (II) shows staining of CD11/CD18 and CD41 are still not discernible. Supplemental Figure 1d and 1e should be added to the main figure to bring additional clarity.

4. Thank you for providing the agarose gel images of the TLR3 siRNA knockdowns. Please add these to the supplemental data.

7. PLOS authors have the option to publish the peer review history of their article (what does this mean?). If published, this will include your full peer review and any attached files.

Reviewer #1: **Yes: **Nithya Mariappan

Reviewer #2: No

Reviewer #3: No

---

## [Author Response · Author response to Decision Letter 1]

8 Nov 2021

Response to Reviewer-2:

Second Revision – author’s’ response 7th Nov., 2021

All editing in the manuscript is presented as track change yellow.

I am indebted to you for the valuable comments for overall improvement of the manuscript. Accordingly, in this revised paper, I give due weightage to comments and the resultant responses are:

The answer to the Reviewer # 2

Major Comments:

Point: 1

 In Figure 1, quantitation of the eDNA, eRNA as well as SI markers were requested. While the quantitation of eDNA and eRNA were provided in the response letter, the plots were not changed to reflect quantitation. The authors fail to realize the significance of adding quantitation to the graphs in Figure 1. The intent is not to convince the reviewer but the reader. These numbers will serve as a guide to investigators in the field who may be doing similar experiments. The justification for not providing these numbers since it is a basic science study and not a clinical study is baseless and without any merit. In similar basic science studies, the authors have themselves published concentrations of eRNA and eDNA (Refs 1and 13 of the manuscript). They have also published quantitative values for ELISA in these publications that they have authored. Interestingly, the y-axis values in Figure 1a and 1b correspond to the values provided in the response letter and are not ‘fold changes’ as indicated in the plots.

Author Response: I wish to thank reviewer for the comments. We have addressed the comment of the reviewer and provided the bar graphs for quantitative analysis of all the SI markers (Figure 1 (c-g). Specific quantitative ELISA kits were used for this purpose. The plots for eRNA and eDNA have been corrected to represent the quantitation details.

Point: 2

The supplementary figures 1a, 1b and 1c should be shown alongside Figure 5(I) to bring more clarity to the changes. The arrows in the lower and higher magnification images do not match and sometimes they show different regions (Supp Fig 1a: Arrows in Poly I:C and RNase A+AH pairs; Supp Fig 1b: Arrows in Poly I:C and RNase A+AH pairs; Supp Fig 1c: Arrows in Poly I:C and eRNA pairs). The figures in the main body of the manuscript should convey the reported findings without having to go to the supplementary data.

Author Response: I thank the reviewer for the suggestions. We have addressed the concern and necessary correction were made. We have merged the higher magnification images for figure 5 with the existing image data set Figure 5 (b-d). The arrow pointers of all the images have been checked and rectified of errors.

Point: 3

Similarly, the structures within Figure 5 (II) shows staining of CD11/CD18 and CD41 are still not discernible. Supplemental Figure 1d and 1e should be added to the main figure to bring additional clarity.

Author Response: I appreciate the reviewer for the comments. Accordingly, we have addressed the suggestion of the reviewer and merged the higher magnification images for figure 5 with the existing image data set Figure 5 (e-f).

Point: 4:

Thank you for providing the agarose gel images of the TLR3 siRNA knockdowns. Please add these to the supplemental data. The background should clearly state the purpose of the study.

Author Response: Acknowledging the comment by the reviewer, we have included the agarose gel image as Supplementary figure 2.

---

## [Editor Report · Decision Letter 2]

18 Nov 2021

Key role of Extracellular RNA in hypoxic stress induced myocardial injury

PONE-D-21-14652R2

Dear Dr. Khan,

We’re pleased to inform you that your manuscript has been judged scientifically suitable for publication and will be formally accepted for publication once it meets all outstanding technical requirements.

Kind regards,

Aftab Ahmad, Ph.D.

Academic Editor

PLOS ONE
---

## [Editor Report · Acceptance letter]

1 Dec 2021

PONE-D-21-14652R2 

Key role of Extracellular RNA in hypoxic stress induced myocardial injury 

Dear Dr. Khan:

I'm pleased to inform you that your manuscript has been deemed suitable for publication in PLOS ONE. Congratulations! Your manuscript is now with our production department. 

Kind regards, 

on behalf of

Dr. Aftab Ahmad 

Academic Editor

PLOS ONE